# The Effect of Agarose on 3D Bioprinting

**DOI:** 10.3390/polym13224028

**Published:** 2021-11-21

**Authors:** Chi Gong, Zhiyuan Kong, Xiaohong Wang

**Affiliations:** 1Center of 3D Printing & Organ Manufacturing, School of Inteligent Medicine, China Medical University (CMU), No. 77 Puhe Road, Shenyang North New Area, Shenyang 110122, China; 2019121147@stu.cmu.edu.cn; 2Center of Organ Manufacturing, Department of Mechanical Engineering, Tsinghua University, Beijing 100084, China; 2020121772@stu.cmu.edu.cn

**Keywords:** 3D bioprinting, agarose, pregelatinization, organ manufacturing

## Abstract

In three-dimensional (3D) bioprinting, the accuracy, stability, and mechanical properties of the formed structure are very important to the overall composition and internal structure of the complex organ. In traditional 3D bioprinting, low-temperature gelatinization of gelatin is often used to construct complex tissues and organs. However, the hydrosol relies too much on the concentration of gelatin and has limited formation accuracy and stability. In this study, we take advantage of the physical crosslinking of agarose at 35–40 °C to replace the single pregelatinization effect of gelatin in 3D bioprinting, and printing composite gelatin/alginate/agarose hydrogels at two temperatures, i.e., 10 °C and 24 °C, respectively. After in-depth research, we find that the structures manufactured by the pregelatinization method of agarose are significantly more accurate, more stable, and harder than those pregelatined by gelatin. We believe that this research holds the potential to be widely used in the future organ manufacturing fields with high structural accuracy and stability.

## 1. Introduction

The concept, connotation and extension of organ manufacturing were firstly defined by Professor Wang in 2003, and it is an emerging discipline combining biology, chemistry, computer, mechanics and materials to construct bioartificial organs in vitro through automatic or semi-automatic methods [1,2]. However, organ manufacturing still has many technical problems to solve. For instance, it is difficult to construct the hierarchical vascular networks in a human organ. To solve these problems, an industrial manufacturing technology that has nothing to do with life science has appeared in people’s vision: three-dimensional (3D) bioprinting. 3D bioprinting technologies have expanded rapidly and overturned the concepts of traditional tissue engineering totally for organ manufacturing during the last decade [3,4,5,6,7].

3D bioprinting is defined as the simultaneous stacking of living cells along with other biomaterials in a prescribed layer-by-layer manner under the instruction of a digital model [8,9]. Recently, organ manufacturing has become a hot topic in research. The reason is that there are too many patients in hospitals who need organ transplantation, and the donor organs are too few. The prospect and profit of organ transplantation are huge. 3D bioprinting technologies provide such capabilities as printing multiple cell types along with different biomaterials under the instruction of the patient’s failure organ model [10,11]. Patients’ own data can be read using medical imaging such as magnetic resonance imaging (MRI) and computed tomography (CT) to build patient-specific implants [12,13,14]. Using this technology, one can produce bioartificial organs suitable for implantation and other purposes quickly and accurately, either on a large scale or customized, thereby saving more lives.

3D bioprinting is based on additive manufacturing technology and allows controlled fabrication of 3D structures in all directions [15,16]. Over the last decade, with the gradual improvement of 3D bioprinting technology, biomaterials have been manufactured into various implants [17]. In particular, stem cells can be printed with polymeric hydrogels to aid tissue/organ repair. So far, mesenchymal stem cells (MSC), embryonic stem cells (ESC) and induced pluripotent stem cells (iPSC) have been used in 3D bioprinting [18]. Meanwhile, most primary cells, such as myocardial cells, are difficult to obtain, and the passage times are limited [19]. Using hydrogels capable of being loaded with cells, 3D bioprinting can accurately construct functional tissues and organs with complex structures [20]. Owing to the possibility of rapidly creating new prototypes and the controllability of material preparation, 3D bioprinting has great potential in biomedical fields [21,22].

The key to printing living tissues/organs is the selection of appropriate cell types and supporting materials, with the ultimate goal of creating physiologically relevant micro, meso and macro cell survival environments with appropriate physical, chemical and biological properties [23,24,25,26,27,28]. The growth and proliferation of cells in the human body cannot be separated from the extracellular matrices (ECMs). Surprisingly, the polymers used for 3D bioprinting can mimic the natural ECMs in human organs with suitable mechanical, hydration, and biological properties [29,30,31,32,33,34,35,36].

Currently, there are three main 3D bioprinting technologies, including material extrusion bioprinting, material jetting bioprinting and vat polymerization bioprinting. Among these technologies, material extrusion bioprinting is the most widely used technology in organ manufacturing areas, because it has the following advantages: great operability, low cost, and small side effects on cells [37]. Material jetting printing uses a fine nozzle to print fluid materials through jetting. The printing accuracy is low because the ejected liquid can be spattered to a certain degree [9]. Vat polymerization bioprinting uses photopolymerization to cure the liquid ink hosted in a vat into a volumetric construct in a layer-by-layer manner. Limitations exist for this technology in organ manufacturing because it is hard to print multiple biomaterials with necessary accuracy [38,39,40].

In material extrusion bioprinting, polymer solutions with or without cells are often selected as printing matrices or “bioinks” because these materials have good biocompatibilities and degradabilities. The polymer solutions need to be in the sol state in the extrusion tube cavity, and in the gel state when they are extruded from the nozzle, because the sol state is maintained in the extrusion tube cavity for the flow of the polymeric materials, and the gel state is maintained in the nozzle for the formation of solid structures. However merely meeting this condition is not enough, before the polymer solutions are placed in the extrusion tube cavity, some of them need to be physically or chemically crosslinked into the gel state. The purpose of this arrangement is to prevent the extruded polymer materials from flowing like liquid and to facilitate material shaping.

Thus, polymers, which are the main components of hydrogels, play a key role in 3D organ printing. Most cell-laden polymer hydrogels are formed through physical, chemical or biochemical crosslinking of homopolymer or copolymer solutions. The behavior of cells in the hydrogels can be controlled by changing the physical and chemical properties of the polymers used. Hydrogels used for 3D organ printing include natural and synthetic polymers, and their combinations. Natural polymer chains are usually full of biologically active groups, which can provide a benign and stable environment for cells, especially stem cells, to grow, migrate, proliferate and/or differentiate [41,42]. Meanwhile, synthetic polymer network is often composed of repeatable inert units. They are generally superior to natural polymers in terms of mechanical properties and immunogenic response.

In 3D bioprinting, there is also the phenomenon of shear thinning (the viscosity decreases with the increase of shear rate or shear stress). The phenomenon of shear thinning is widespread in polymer fluids, which makes it difficult to control the extruded hydrogel, which in turn affects the accuracy and stability of the structure. So far, this problem still plagues the majority of researchers. Even though shear thinning behavior assessed by steady-state shear measurements (viscosity as function of shear rate) suggests the possibility of injecting the material through a needle, functional injectability tests clearly demonstrate the injectability through needles with specific diameter providing information on the maximum load (and hence pressure) involved in the process, which should be consequently optimized according to the obtained results. In addition, the viscoelastic properties before and after the injection through the employed nozzles are important for the process and for the final properties of the fabricated devices (e.g., the injection through nozzles may strongly alter the material properties).

Gelatin, a natural polymer with great biocompatibility, is selected as the pregelatinized matrix in most of the traditional extrusion-based bioprinting technologies, because gelatin can be physically crosslinked at low temperature (e.g., 10–30 °C) [29,30,31,32,33,34,35,36]. However, the lower the temperature is, the greater the damage to the cells is. Since gelatin is highly sensitive to temperature, under conditions where the molding chamber does not meet the requirements of high-precision temperature control, the material deposition often does not have high structural precision and stability.

In the present study, we develop a new kind of polymeric material that can substitute the pregelatinization process of gelatin. With this polymeric material, we have broken through the shortcoming of the traditional gelatin-based “bioinks”, in which the sol–gel state transition process is totally dependent on temperature-sensitive gelatin.

## 2. Materials and Methods

### 2.1. Materials

Gelatin (CAS: 9000-70-8, Glue strength ~100 g Bloom) was purchased from Aladdin, Shanghai, China. Sodium alginate (CAS: 9005-38-3) was purchased from Sigma, St. Louis, MO, USA. Agarose (CAS: 9012-36-6) was purchased from Adamas, Emeryville, CA, USA. Transglutaminase (TG) (100 U/g, Dongsheng, Shanghai, China) was purchased from Dongsheng. Other reagents used in the experiments were all of an analytical grade and purchased from Beyotime Biotechnology (Shanghai, China).

### 2.2. Preparation of Gelatin-Alginate-Agarose Mixtures

Gelatin, sodium alginate and agarose were dissolved in phosphate buffered saline (PH = 7.4), respectively, to form solutions with different concentrations, 45% (*w*/*v*) for gelatin, 6% (*w*/*v*) for sodium alginate, and 0.6% (*w*/*v*) for agarose. The three solutions were combined at a volume ratio of 1:1:1 to form a 17.4% (*w*/*v*) solution (G15S2A0.2) at 90 °C before the PH was adjusted to 7.3 and kept at 50 °C for 1 h.

### 2.3. Cells Acquisition

Adipose-derived stem cells (ASCs) are a type of pluripotent stem cells in adipose tissue. Recently, ASCs have attracted great attention because they have a wide range of sources, are easy to obtain, do not involve ethical issues, are convenient for autologous transplantation, and bring less pain to patients. ASCs have been induced to differentiate into many cell types, such as endothelial cells, osteoblasts, chondrocytes, fibroblasts under in vivo and in vitro conditions.

ASCs were isolated from 8-week-old Sprague-Dawley (SD) male rats by the method described elsewhere. Briefly, the rats were sacrificed, and the inguinal adipose tissue was isolated, washed and cut into chyliform. Then the adipose tissue was digested with 1 mg mL collagenase II (2200929, Gibco) diluted in DMEM/F-12 (AF29584968, Cytiva) at 37 °C for 60 min. Next, the media was filtered with 70-μm filter screen, and the digestion was stopped by adding equal amount of culture media. The media was centrifuged, and the supernatant discarded. Then culture media was added, and the centrifugation was repeated again. Finally, the harvested cells were cultured in cell incubator (37 °C, 5% CO_2_). Cells at passage 1–4 were used for the subsequent experiments.

### 2.4. Preparation of Cell-Laden Hydrogels

ASCs were embedded in the gelatin–alginate–agarose solution at a density of 1 × 10^7^ cells/mL.

### 2.5. Manufacturing and Optimizing Procedures

The manufacturing process is briefly described as follows: after the ASCs cells were mixed with the gelatin–alginate–agarose solution at a density of 1 × 10^7^ cells/mL, the mixed solution was poured into the printing needle tube for pregelatinization at a certain temperature before being printed. The 3D extrusion bioprinting parameters had been set in Table 1.

### 2.6. Cell Viability and Proliferation Rate Assays

Cell viability assay was performed using a CCK-8 kit (Vazyme, Nanjing, China) following the instructions. ASCs were encapsulated in the designed hydrosols with a density of 2 × 10^7^ cells/mL. After the cell-laden hydrosols were printed, the 3D constructs were chemically crosslinked using 1% CaCl_2_ solution. A piece of the 3D construct was cut off and stained with CCK-8 kit (100 µL medium + 20 µL CCK-8) [43].

After staining, the solution was transferred to a blank 96-well plate to detect the optical density (OD) at 450 nm exciting light (Thermo Fisher Scientific, Waltham, MA, USA). The mean OD values of the groups were expressed as OD_t_, while the control groups were expressed as OD_n_. Cell viability (CV) was calculated via the following formula.
(1)$$\mathrm{CV}=\frac{{\mathrm{OD}}_{t}-{\mathrm{OD}}_{n}}{{\mathrm{OD}}_{n}}$$

Each experiment was performed in four replicas.

### 2.7. Characterization of Cell Survival States

Cell survival states were assessed using an acridine orange (AO)/propidium iodide (PI) double staining kit, i.e., fluorescent live/dead viability assay kit (BestBio, Beijing, China), according to the instructions. The samples were first immersed in PBS containing 5 µL of AO and 10 µL of PI and incubated in dark at 4 °C for 10 min. After being washed with PBS three times, they were checked using a laser confocal microscope (LSM, N1R, Nikon, Japan) at 488 nm exciting light. Dead cells were stained a red color while living cells were stained a green color.

Fluorescent live/dead viability assay kit (Biosharp, CAS:28718-90-3) was employed, according to the instructions. 2-(4-Amidinophenyl)-6-indolecarbamidine dihydrochloride (DAPI) solution, a fluorescent dye that can bind strongly to DNA, was used for fluorescence microscopy observations. Because DAPI can penetrate intact cell membranes, it can be used to distinguish live and dead cells. A small piece of the 3D printed construct was stained. After standing at room temperature for 5 min, the DAPI dye solution was aspirated and washed 3 times using PBS, the samples were observed under the fluorescence microscope.

### 2.8. Microscopy Examination

Based on the fact that the concentration of gelatin has the greatest impact on the microscopic pore structure of the materials, we only chose gelatin as a variable for the microscopic observation of the materials. Four concentrations of gelatin-based composite hydrogels were used as described in Table 2. A certain amount of 10% (*w*/*v*) transglutaminase (TG) and 2% (*w*/*v*) CaCl_2_ solution were used to crosslink the polymer chains. After dehydration, the samples were freeze-dried and inspected using a scanning electron microscope (SEM) (JSM-7001F, Tokyo, Japan).

### 2.9. Comparison of Mechanical Properties of the Four Composite Hydrogels

The mechanical properties of hydrogels play a vital role during the 3D printing process and tissue/organ formation stages, and the hardness is the most intuitive characteristic of the mechanical properties. In particular, different polymer concentrations in the hydrogel have the greatest impact on the hardness. For this reason, we tested the hardness of four groups of hydrogels with different polymer ratios, as shown in Table 2.

The test process was as follows: a quick-permeable cylinder mold with a diameter of 10 mm and a height of 20 mm was prepared using filter paper. After the composite polymer solutions were poured into the mold, a 2% (*w*/*v*) CaCl_2_ solution was added to crosslink the sodium alginate, and a 10% concentration of transglutaminase (TG) solution was added to crosslink the gelatin.

A Shore hardness tester HT-6510OO was used to test the hardness of the cylinder hydrogels. When the handle was slowly pressed down by 10 mm, the maximum number was recorded.

### 2.10. Comparison of Water Holding Capacity of the Four Composite Hydrogels

Hydrogels have the characteristics of being able to absorb a large amount of water, and exploring the water-holding properties of matrix materials with different ratios has important implications for 3D bioprinting and subsequent cell proliferation.

Polymer solutions with 15% (*w*/*v*) gelatin, 2% (*w*/*v*) sodium alginate and 0.2% (*w*/*v*) agarose, 0%, 5%, 10%, 15%, 20% of gelatin, 0%, 2%, 4%, 6%, 8% of sodium alginate, and 0%, 0.2%, 0.4%, 0.6%, 0.8% of agarose mixed materials were prepared, respectively. Hydrogel samples were prepared according to the method described above. After crosslinking with 2% (*w*/*v*) CaCl_2_ and 10% (*w*/*v*) TG solutions, water holding capabilities (WHCs) of the samples were calculated according to the following formula (2). After polymer took shape, the samples were washed with Hepes buffer for 10 min, before being weighed using an electronic balance to determine the wet weight (M_w_). Dry weight (M_d_) was recorded after freeze-drying.
(2)$$\mathrm{WHC}=(M_{w}-M_{d})/M_{d}$$

### 2.11. 3D Bioprinting with Cells

The cell-laden hydrogels with 15% (*w*/*v*) gelatin, 2% (*w*/*v*) sodium alginate, and 0.2% (*w*/*v*) agarose prepared in Section 2.6 were printed at 10 °C. A needle with an inner diameter of 0.573 was used with a solid cylinder model similar to a blood vessel. After a 5-layer cylindrical cell-containing structure was obtained, it was immersed in F-12 culture medium for in vitro cultures. The medium was changed once a day, and a piece of the samples was cut every two days for DAPI staining and observation under a fluorescence microscope.

### 2.12. Statistic Analysis

Sample values are expressed as means ± standard deviation (SD). Values were analyzed using analysis of variance (ANOVA). A *p* value of <0.05 was considered significant. SPSS 13.0 statistic software was used.

## 3. Results

### 3.1. Morphologies of Hydrogels with Different Polymer Ratios

The micropores in the hydrogels have two important roles for 3D organ printing: (1) The pores that are equivalent to or slightly larger than the cells, about 1~20 μm, can provide attachment support for cell growth, regulate the morphology of cells, and provide microchannels for nutrient diffusion; and (2) larger pores, about 20~100 μm, can provide enough space for stem cell proliferation and tissue growth, as well as larger channels for nutrient diffusion.

In traditional 3D bioprinting, the physical crosslinking properties of gelatin at low temperatures (e.g., 10–30 °C) are often used to complete the transformation from hydrosol to hydrogel. As presented in Figure 1a–c). As the concentration of gelatin gets higher and higher, the pores in the hydrogels become smaller and smaller. In particular, when the concentration of gelatin reaches 20%, many pores disappear (Figure 1c). The pore structure is the living space of cells. When the living space of cells decreases, it is difficult for cells to grow and proliferate rapidly. In this respect, we conclude that gelatin content that is too high is not beneficial for cell growth in the hydrogels.

Studies have shown that the carboxyl group of alginate can combine with the amino group of gelatin to form a polyelectrolyte complex dominated by electrostatic attraction [44]. Therefore, the mechanical strength of the hydrogel composed of gelatin and sodium alginate is stronger than that of pure gelatin, and the sodium alginate can combine with divalent cations to transform from sol to gel state, which can improve the stability of the hydrogel. As presented in Figure 1d, the composite hydrogel without sodium alginate has a large number of pores, and the diameter of the pores is small. As presented in Figure 1e, when the concentration of sodium alginate is 4%, the pore size in the hydrogel is obviously larger, but the number of pores is also reduced. The reason is that the combination of gelatin and sodium alginate reduces the mobility between molecules. As presented in Figure 1f, the pore size in the hydrogel did not continue to increase, but some pores were filled with excess polymers, reducing the number of pores. Considering the stability of the composite hydrogels and the porosity of the mixed materials, we believe that the optimal concentration of sodium alginate should be between 0% and 4%.

Compared with gelatin and sodium alginate, agarose is significantly more stable. As presented in Figure 1g, the pores of the hydrogel without agarose are smaller, some areas are still filled with polymer molecules. However, as shown in Figure 1h, when the concentration of agarose was only 0.4%, the pores of the hydrogel became larger, and the number also became larger. When the concentration of agarose was 0.8%, the microstructure of the hydrogel did not change much except for a small decrease in the number of pores (Figure 1i). This shows that adding a certain amount of agarose to the hydrogel can increase the size and number of pores to a certain extent, which may be beneficial to the growth and proliferation of embedded cells.

According to the appearance of the microstructure of the matrix with different material components (Figure 2), we constructed Table 2, in order to more accurately quantify the average matrix wall thickness, pore diameter and porosity with different material components.

According to Figure 2 and Table 2, we can conclude the following: a certain amount of gelatin increases the average matrix wall thickness, reduces the pore diameter, and reduces the porosity; a certain amount of alginate increases the average matrix wall thickness, increases the pore diameter, and reduces the porosity; and a certain amount of agarose increases the average matrix wall thickness, increases the pore diameter, and increases the porosity.

### 3.2. Mechanical Properties of Different Hydrogels

It is well known that the ASTM D2240-00 testing standard calls for different scales. There are three measurement units for hardness: (OO), (A) and (D). They have their own standard hardness testers. (OO) is usually used to measure materials that are softer than tofu. The hydrogel material we used to measure is right in line with (OO) measurement standards.

Figure 3a shows that when the concentration of gelatin increases, the hardness of the hydrogels also increases, implying that gelatin can improve the mechanical properties of the hydrogels. Figure 3b shows that when the concentration of sodium alginate increases, the hardness of the hydrogel also increases. Similarly, when the concentration of agarose increases, the hardness of the hydrogel also increases (Figure 3c). This means that all the molecules of gelatin, sodium alginate, and agarose can increase the mechanical properties of the composite hydrogels.

It can be seen from Figure 3a that the hardness of pure gelatin is obviously different from the hardness of gelatin-based composite hydrogels. The hardness of the pure sodium alginate is smaller than that of the sodium alginate-based composite hydrogels (Figure 3b).

In the composite hydrogels, gelatin is a major component, and plays a special role during the 3D printing processes with the sol–gel transformation. After CaCl_2_ chemical crosslinking, the mechanical strength of the composite hydrogels mainly depends on the sodium alginate molecules. The COO-carboxyl groups of sodium alginate can form polyelectrolyte complexes with the main static electricity of NH3+ amino groups of gelatin, so the mechanical strength of the combined gelatin/sodium is obviously stronger than those of the two single polymers. Agarose has a significant effect on the overall mechanical properties of the composite hydrogels in the range of 0% to 0.35%, but after agarose exceeds this range, the mechanical properties of the composite hydrogels change slightly. The reason for this may be that agarose is a linear polymer with limited strength and cannot be crosslinked with other molecules. If the concentration is too high, it can directly affect the binding effects between gelatin and sodium alginate molecules. Moreover, when the concentration of agarose is high enough, the polymer solutions can automatically translate into solid state at room temperature, which is not beneficial for the 3D bioprinting process.

### 3.3. Water Holding Capacity of the Hydrogels

As shown in Figure 4, when the concentration of gelatin increases, the water holding rate of the composite hydrogels shows an acute downward trend. Meanwhile, when the concentration of sodium alginate increases, the water holding capacity of the composite hydrogels also decreases sharply. However, when the concentration of agarose increases, the water holding capacity of the composite hydrogels decreases slightly at first, and when the concentration is greater than 0.4%, the water holding capacity rises slightly.

### 3.4. The Formation Quality of Gelatin–Alginate–Agarose Hydrogels at 10 °C

The ratio of the composite hydrogel was 15% gelatin, 2% sodium alginate and 0.2% agarose. As presented in Figure 5, the entire 3D bioprinting process was extremely stable, continuous and precise. Though there were some little flaws from Figure 5a–c, the phenomenon of uneven or incapable extrusion of materials during the bioprinting process was greatly reduced. Compared with our former pure gelatin controlled bioprinting, the height limitation (e.g., 2 × 2× 2 cm^3^) was overcome [1]. An astonishing 113-layer (Figure 5d) construct was achieved, setting a precedent for the manufacturing of large-scale and complex organs in the future.

### 3.5. The Formation Quality of Gelatin–Alginate–Agarose Hydrogel at 24 °C

The ratio of the composite hydrogel was 15% (*w*/*v*) gelatin, 2% (*w*/*v*) sodium alginate and 0.4% (*w*/*v*) agarose. As presented in Figure 6, the accuracy, stability and continuity of printing were undoubtedly improved compared with structures printed at 10 °C (Figure 5). From Figure 6a–c, the contour of the structures was extremely straight without any flaws on the overall structures. There was almost no nozzle clogging during the entire printing process, the overall forming quality of structures was greatly improved.

### 3.6. The Effect of Extrusion Pressure on Cell Viability

In 3D bioprinting, the extrusion of “bioinks” requires a certain amount of pressure. However, in our experiments, we found that from the beginning of the extrusion operation, the pressure required to extrude the material increased continuously, before finally remaining at a relatively constant value. There are many reasons for this phenomenon, such as the shear thinning of polymer materials, the influence of low temperature of the platform, and so on.

As presented in Table 3, the pressure of traditional temperature-controlled bioprinting technology changes most significantly with time, and the pressure approaches 0.6 MPa, the pressure of the semi-temperature-controlled bioprinting approaches 0.4 MPa, and the pressure of the non-temperature-controlled bioprinting approaches 0.3 MPa. This phenomenon also indirectly shows that the influence of temperature on material extrusion cannot be ignored. The closer the temperature is to room temperature, the lower the pressure required for extrusion.

As presented in Figure 7, The cell viability did not decrease as we expected with increasing extrusion pressure. On the contrary, there was almost no decrease in cell viability at 0 MPa, 0.2 MPa and 0.4 MPa, while the cell viability showed a slight decrease at 0.5 MPa. In addition, 0.5 MPa is basically the same as the highest value of the extrusion pressure in the experiment, so the extrusion pressure of the three 3D bioprinting technologies has an acceptable degree of impact on cell viability, and the technology has the least impact. In addition, Table 4 presents the information depicted Figure 7, serving to more accurately and quantitatively reflect the death rate of cells.

### 3.7. Effect of Agarose Content on Formation Quality

Due to agarose having a strength that cannot be compared to other polymer materials, we considered adding some agarose to maintain the overall shape of the structure when constructing a higher cell-material structure. The structure was placed at room temperature to physically crosslink the agarose in the structure.

As presented in Figure 8, the deformation resistance of the structures tended to increase when the agarose ratio in the hydrogel was increased from 0% to 0.2%, and to 0.4%. In Figure 8a, due to the lack of intense support from agarose, the middle and lower parts of the structure significantly collapsed, and the overall shape of the structure changed. In Figure 8b, the overall order of the structure was greatly improved with the addition of only 0.2% agarose; however, there was a slight collapse at the bottom of the structure. In Figure 8c, the overall shape of the structure was very precise, and there was no deformation or collapse from top to bottom. It is clear that agarose has a very significant effect on the precision and compression performance of the formation of structure.

As expected, when the hydrogel in 3D bioprinting uses the physical crosslinking of agarose instead of the physical crosslinking of gelatin to shape the structure, the stability of the structure is greatly improved. This is because the degree of crosslinking of gelatin is not complete when the temperature is below 30 °C. A small change in temperature will cause the degree of crosslinking of gelatin to change. However, the agarose is completely crosslinked below 30 °C, so the stability of the hydrogel is not affected by temperature. When extruding, once the stability of the hydrogel is improved, the accuracy of printing is naturally improved.

### 3.8. Effect of Formation Time on Cell Viability

As is known, the temperature of the forming chamber will have a salient influence on the survival rate of cells. When the temperature is lower than 37 °C, the survival rate of cells will decrease to a certain extent. However, these two new bioprinting methods can greatly increase the temperature of the forming chamber and thus increase the survival rate of cells.

Since the temperature of the controlled technology in the molding chamber are 4 °C, 10 °C and room temperature, respectively, we used the CCK8 method to measure the cell survival rate of 3D bioprinting at different times at these three temperatures.

As presented in Figure 9, cell survival rate of the samples with the temperature of 4 °C, 10 °C and 24 °C were 91%, 94% and 98%, respectively, when the formation time was 0.5 h. Then, when the formation time was 1 h, the cell survival rate of 4 °C remained 87%, the cell survival rate of 10 °C remains 90% and the cell survival rate of 24 °C still remained 96%. When the formation time reached 2 h, the cell survival rate of 4°C remained 83%, the cell survival rate of 10 °C also remained 86% and the cell survival rate of 24 °C still remained 93%. Within 2 h, the difference between these three processing temperatures was still not significant enough. However, when the formation time reached 5 h, the cell survival rate at these three formation temperatures was already significantly different. The cell survival rate of the samples with the temperature of 4 °C, 10 °C and 24 °C were 70%, 77% and 89% respectively when the formation time was 5 h.

### 3.9. Formation of Vascular Structures under Two Printing Methods

As presented at Figure 10, The forming lines of non-temperature-controlled bioprinting are rough, but the forming lines of semi-temperature-controlled bioprinting are precise. This is because a certain degree of temperature control can better ensure that the printing material is a liquid before extrusion and a solid after extrusion. The printing materials without temperature control are required to have a certain degree of fluidity and a certain degree of formability. To achieve these two characteristics, the printing material must always maintain a solid–liquid mixed state, and the formation is extremely dependent on the assistance of a crosslinking agent. Therefore, this determines that the forming accuracy of non-temperature-controlled bioprinting is not as good as that of semi-temperature-controlled bioprinting.

However, as shown in Figure 10b, the inside of the blood vessel-like structure under semi-temperature-controlled bioprinting shows a certain degree of collapse, while the inside of the blood vessel-like structure under the non-temperature-controlled bioprinting maintains its original shape. This is because the forming temperature of non-temperature-controlled bioprinting is normal temperature, and it is not easily affected by external temperature to cause deformation of the structure, while the forming temperature of semi-temperature-controlled bioprinting is 10 °C, which is easily affected by external temperature. This influence leads to the change of the shape of the structure. All in all, this difference in forming temperature is the key to determining whether the overall shape of the material can be effectively maintained.

In terms of accuracy, there are many determinants, among which stability and resolution are very critical factors. Generally speaking, the higher the stability, the higher the resolution, but the opposite result was obtained in this experiment. The main reason is that the material must complete the conversion from liquid to solid during extrusion, and the non-temperature-controlled bioprinting must rely on the auxiliary crosslinking of the crosslinking agent without temperature control. This also shows that the use of temperature to control the solid–liquid transition of materials is still the most efficient method so far.

### 3.10. Construction of Trigeminal Branch Blood Vessels

In 3D bioprinting, if you want to build complex tissues and organs, you must have higher requirements for the precision of the forming. Therefore, we chose the agarose semi-temperature-controlled technology with higher precision to construct the trigeminal vessel-like structure.

To verify the feasibility of the trigeminal vessel-like structure, we first tried to print a solid trigeminal vessel-like structure with stronger stability. As presented in Figure 11a,b, the solid trigeminal vessel-like structure (vessel diameter of 15mm) that we printed out conforms to the trigeminal vessel. However, due to the small size of the structure and the thicker extruded material, the pores in the top view are not obvious, but the contour lines are still faintly visible.

After verifying the feasibility of forming a solid trigeminal vessel-like structure, we designed a hollow trigeminal vessel-like model (blood vessels with an outer diameter of 20 mm and an inner diameter of 8 mm), as shown in Figure 11c. However, in the process of realizing the formation of the hollow trident vascular structure, we found that the existing methods can no longer satisfy the stability of material extrusion. This is mainly because the size of the hollow trigeminal vascular structure exceeds that of the previous structure, and the forming time (3 h) is too long, which puts forward higher requirements for the stability of the material. As presented in Figure 11d, To maintain fluidity in the syringe, the material used for extrusion must be in a sol state in the syringe; and in order to have a certain formability after extrusion, it must be in a gel state on the platform. The problem we often encounter when constructing large structures is that the state of the material in the syringe is in a gel state. However, we only control the temperature of the platform. This is acceptable for building smaller structures, but not when building larger structures.

To solve this problem, we set up a simple heating device on the periphery of the syringe, as shown in Figure 11e. The heating plate maintains a temperature of 35 °C. From the actual forming situation, the stability of the extruded material is greatly improved.

As presented in Figure 11f–h, The shape of the structure roughly meets the appearance of a trigeminal blood vessel, without obvious deformation, and the pores of the structure basically meet the necessary survival conditions for cells. The internal blood vessels are unblocked, which is conducive to blood circulation.

### 3.11. The Results of 3D Bioprinting with Cells

As presented in Figure 12a, we printed the cell-containing hydrogel on a petri dish, and the printed shape was cylindrical. Then we added the F-12 broth containing the crosslinking agent (MTG enzyme solution and calcium chloride solution) to the petri dish as shown in Figure 12b.

After a day of cultivation, we observed the scene as shown in Figure 12c under a fluorescence microscope. Since the cell concentration in the cell-containing hydrogel we prepared was only 7.5 × 10^6^ per milliliter, there were not many cells observed under the fluorescence microscope. As presented in Figure 12d, after two days of culture, observation under a fluorescence microscope clearly showed that the number of cells had increased slightly, which indicates that the material had better biocompatibility. After culturing for four days, the number of cells had increased to a certain extent as presented in Figure 12e. Time went by little by little. On the sixth day, the cells had spread all over the area under the fluorescence microscope, as presented in Figure 12f. The changes of cells from day one to day six fully illustrate that in 3D bioprinting, the hydrogel we formulated has good biocompatibility, and cells can survive and proliferate. It has good feasibility that the technology is used to construct complex tissues. In addition, Table 5 presents the information depicted in Figure 12c–f, in order to more accurately and quantitatively reflect the death rate of cells.

Cell proliferation rate in the hydrogel for printing is characterized using a CCK-8 kit on day 1, 2, 3, and 5. As presented in Figure 13, the cell viability increases constantly without a plateau during the 5 days of in vitro cultures. This result is consistent with the images shown in Figure 12, where ASCs cells are all in living states (blue) with augmented aggregates over time. However, compared with low-concentration hydrogels, high-concentration hydrogels still have a certain inhibitory effect on the proliferation and differentiation of cells, which can also be visually expressed from the statistically significant marks in Figure 13. Further research is still in progress.

## 4. Discussion

In this study, we mainly studied the role of agarose in 3D bioprinting, and used the characteristic of the physical crosslinking of agarose at 35–40 °C to replace the single pregelatinization effect of gelatin in 3D bioprinting. Composite gelatin–alginate–agarose hydrogels were printed at two temperatures, i.e., 10 °C and 24 °C, respectively.

Agarose is generally heated to above 90 °C to dissolve in water, and a good semi-solid gel forms when the temperature drops to 35 °C to 40 °C. Its gelatinicity is caused by the existence of hydrogen bonds. Agarose is hydrophilic, and there are almost no charged groups in its molecules.

When the composite gelatin–alginate–agarose hydrogels are printed at a forming temperature of 10 °C, structural stability relies on the physical crosslinking of part of the gelatin and the physical crosslinking of all agarose to transform the hydrosol to a hydrogel. The advantage is that the formation accuracy is high, but the disadvantage is that the stability after forming is poor.

When the composite gelatin–alginate–agarose hydrogels are printed at a forming temperature of 24 °C (e.g., room temperature), the physical crosslinking of all agarose is relied on to transform the hydrosol to a hydrogel. The advantage is that the stability after printing is greatly improved, but the disadvantage is that the formation accuracy is not high.

From the image obtained by the scanning electron microscope, as the concentration of gelatin gets higher and higher, the pores in the hydrogels become smaller and smaller in Figure 1a–c. In particular, when the concentration of gelatin reaches 20%, many pores disappear. However, pores are the space for cells to survive, which inevitably affects the proliferation and growth of cells. During traditional printing, a certain amount of gelatin is required to maintain the stability and precision of the structure. Therefore, how to choose the concentration of gelatin to balance the cell survived environment and the stability of the structure has become a key problem in the construction of complex organs.

The combination of sodium alginate and gelatin has natural advantages. When the concentration of sodium alginate is between 0% and 4%, the stability and porosity of the hydrogel are the most suitable. This view can be confirmed by SEM images (Figure 1d–f).

Adding a certain amount of agarose to the hydrogel can increase the size and number of pores in the hydrogel, which is also conducive to the proliferation and growth of cells and the printing of complex organs. This view can also be confirmed from the image obtained by the SEM images (Figure 1g–i).

Comparing the hardness test results of the composite hydrogels, we can easily conclude that when the same concentration of gelatin or sodium alginate or agarose is increased, the hardness of the mixed material is improved by agarose more strongly than by sodium alginate, and sodium alginate is stronger than gelatin (Figure 3).

The test results of water holding rate show that the increase in the concentration of gelatin and sodium alginate cause the water holding rate of the mixed material to decrease, and the concentration of agarose added has the least effect, which may also be related to the number of molecules added to agarose (Figure 4).

The pressure required for agarose-containing hydrogels in bioprinting is significantly reduced, but in the actual observation results of staining of cells, this pressure change has almost no effect on the cells (Figure 7). Considering that the extrusion pressure of bioprinting only varies between 0~0.6 MPa, and the survival of cells under pressure greater than 0.6 MPa has not been measured, it is currently impossible to conclude that extrusion pressure has almost no effect on cells.

Whether it is traditional bioprinting or the bioprinting used in this article, in a short period of time, the temperature causes less damage to cells. However, when the printing time is longer, traditional bioprinting causes more damage to the cells than the bioprinting used in this article (Figure 9). Therefore, the agarose-based bioprinting method studied in this paper has obvious advantages for constructing large-scale tissues and organs.

From the effect of the 3D bioprinting, the printing height of the composite hydrogels with agarose is obviously higher than that of without agarose, and the accuracy is also improved to a certain extent (Figure 8). This also shows to a certain extent that agarose has significance in promoting the construction of complex and large tissues and organs. Although the composition of the 3D bioprinted bioinks and culture medium has not been well optimized and standardized in different laboratories and studies, we believe that our in-depth research on agarose will provide a reference for the standardization of bioinks.

## 5. Conclusions

In 3D bioprinting, the addition of a certain agarose can greatly increase the hardness of the structure, and the precision of the print can be improved, and even the temperature set in the forming chamber can be changed. The properties of agarose in the hydrogels are confirmed through physiochemical and biochemical characterizations, such as optical microscope and SEM images, WHCs, and hardness results. The microscopic pores of agarose-containing hydrogels are suitable for cell survival and have good water-holding properties. The proper formation temperature is advantageous to cell survival and reduces the pressure required for printing. These improvements have great reference significance for the selection of polymers and ratio of hydrogel materials. The survival of the cells in the structure was excellent, which fully illustrates their amazing biocompatibility. A great choice is provided when constructing large complex organs with 3D biological printing techniques.

## Figures and Tables

**Figure 1 polymers-13-04028-f001:**
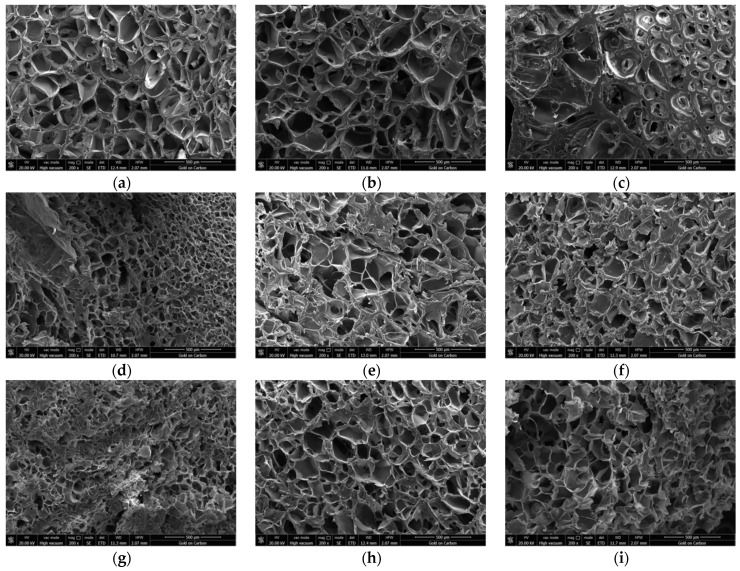
Scanning electron microscopy (SEM) micrographs of the alginate/gelatin/agarose hydrogels. (**a**) Hydrogel composed of 10% gelatin, 2% sodium alginate and 0.2% agarose. (**b**) Hydrogel composed of 15% gelatin, 2% sodium alginate and 0.2% agarose. (**c**) Hydrogel composed of 20% gelatin, 2% sodium alginate and 0.2% agarose. (**d**) Hydrogel composed of 5% gelatin, 0% sodium alginate and 0.2% agarose. (**e**) Hydrogel composed of 5% gelatin, 4% sodium alginate and 0.2% agarose. (**f**) Hydrogel composed of 5% gelatin, 8% sodium alginate and 0.2% agarose. (**g**) Hydrogel composed of 5% gelatin, 2% sodium alginate and 0% agarose. (**h**) Hydrogel composed of 5% gelatin, 2% sodium alginate and 0.4% agarose. (**i**) Hydrogel composed of 5% gelatin, 2% sodium alginate and 0.8% agarose.

**Figure 2 polymers-13-04028-f002:**
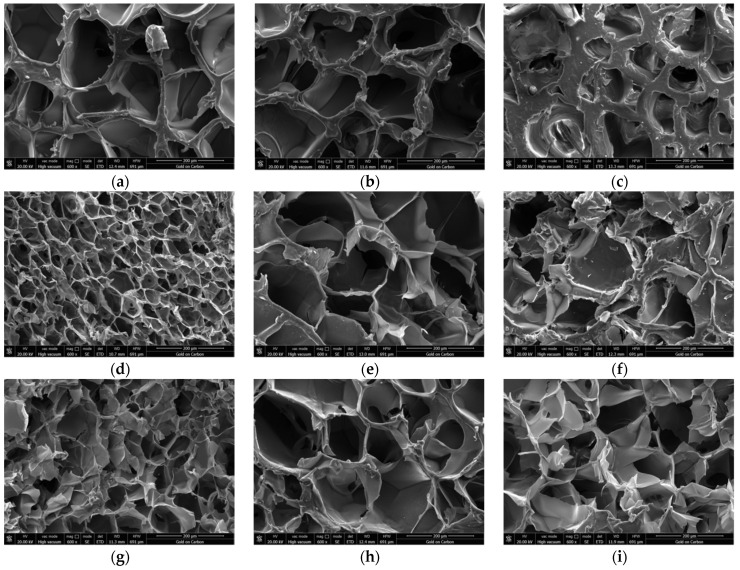
Scanning electron microscopy (SEM) micrographs of the alginate/gelatin/agarose hydrogels. (**a**) A magnified image of hydrogel composed of 10% gelatin, 2% sodium alginate and 0.2% agarose. (**b**) A magnified image of hydrogel composed of 15% gelatin, 2% sodium alginate and 0.2% agarose. (**c**) A magnified image of hydrogel composed of 20% gelatin, 2% sodium alginate and 0.2% agarose. (**d**) A magnified image of hydrogel composed of 5% gelatin, 0% sodium alginate and 0.2% agarose. (**e**) A magnified image of hydrogel composed of 5% gelatin, 4% sodium alginate and 0.2% agarose. (**f**) A magnified image of hydrogel composed of 5% gelatin, 8% sodium alginate and 0.2% agarose. (**g**) A magnified image of hydrogel composed of 5% gelatin, 2% sodium alginate and 0% agarose. (**h**) A magnified image of hydrogel composed of 5% gelatin, 2% sodium alginate and 0.4% agarose. (**i**) A magnified image of hydrogel composed of 5% gelatin, 2% sodium alginate and 0.8% agarose.

**Figure 3 polymers-13-04028-f003:**
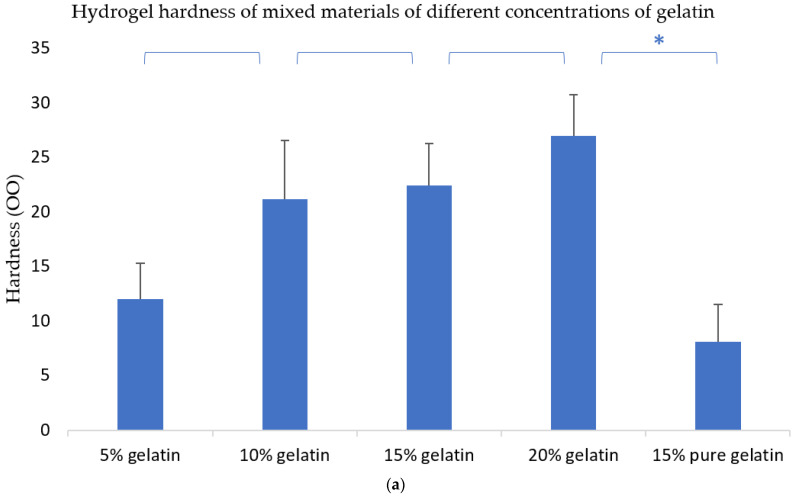
Hardness of the hydrogels. * means that there are statistical significances between the two adjacent gelatin–alginate–agarose concentrations. (**a**) Hydrogel hardness of mixed materials of different concentrations of gelatin. (**b**) Hydrogel hardness of mixed materials of different concentrations of sodium alginate. (**c**) Hydrogel hardness of mixed materials of different concentrations of agarose.

**Figure 4 polymers-13-04028-f004:**
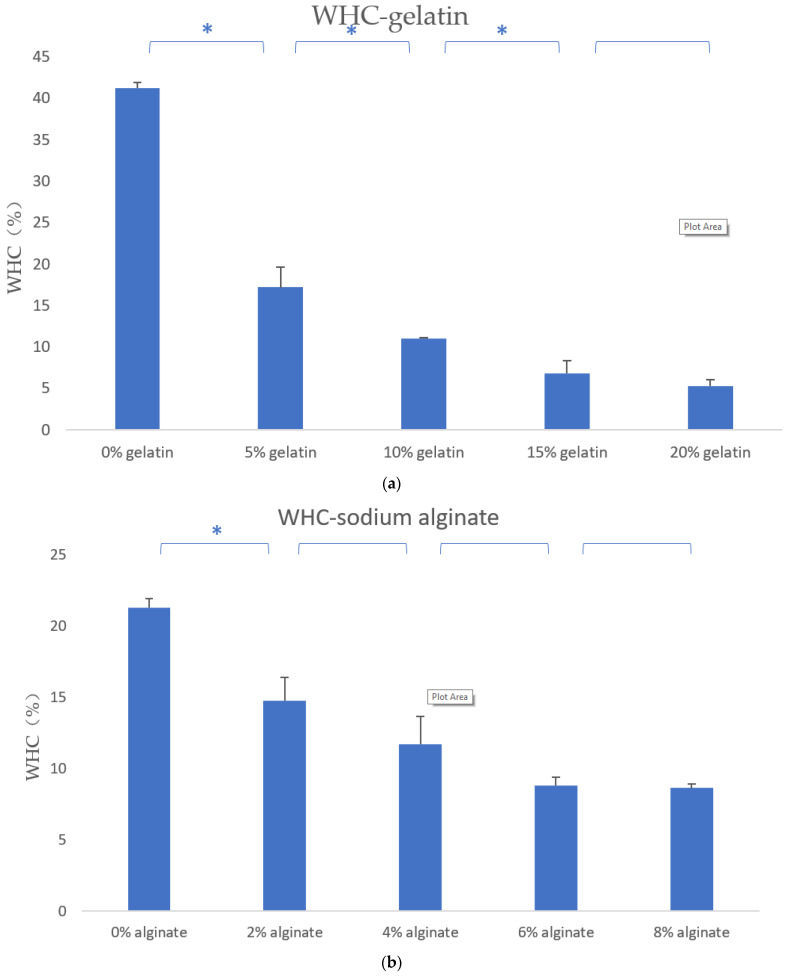
The WHC test results. * means that there are statistical significances between the two adjacent gelatin–alginate–agarose concentrations.

**Figure 5 polymers-13-04028-f005:**
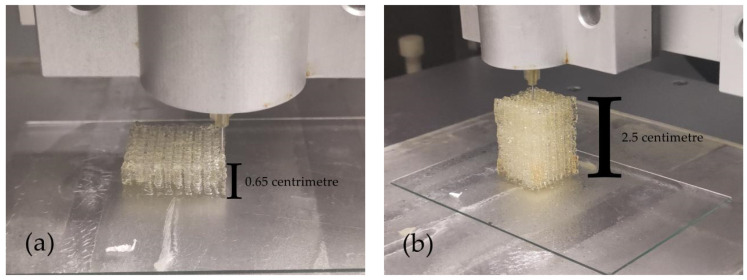
The 3D-printed structures of gelatin/alginate/agarose hydrogel at 10 °C. (**a**) Twenty-layer structure. (**b**) Eighty-layer structure. (**c**) The side top view of one-hundred-and-thirteen-layer structure. (**d**) The front view of one-hundred-and-thirteen-layer structure.

**Figure 6 polymers-13-04028-f006:**
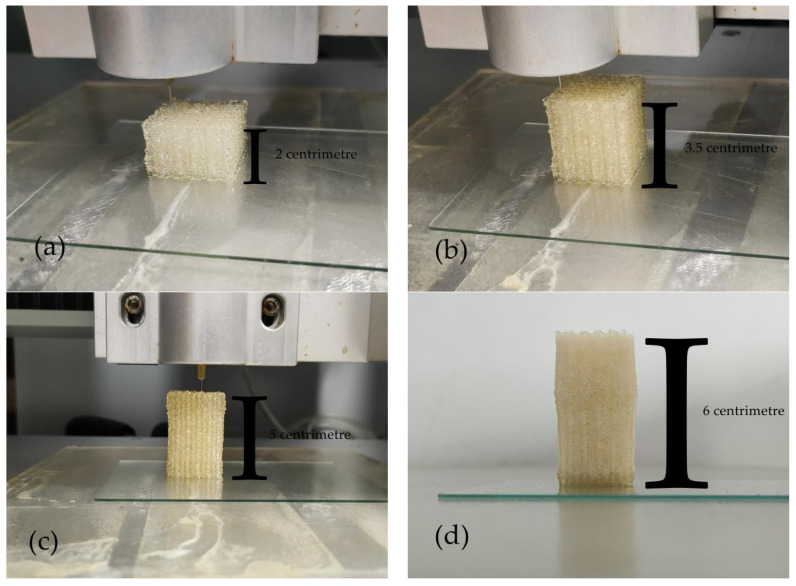
The 3D printed structures of gelatin/alginate/agarose hydrogel at 24 °C. (**a**) A fifty-layer structure. (**b**) An eighty-layer structure. (**c**) A one-hundred-and-ten-layer structure. (**d**) A one-hundred-and-thirty-six-layer structure.

**Figure 7 polymers-13-04028-f007:**
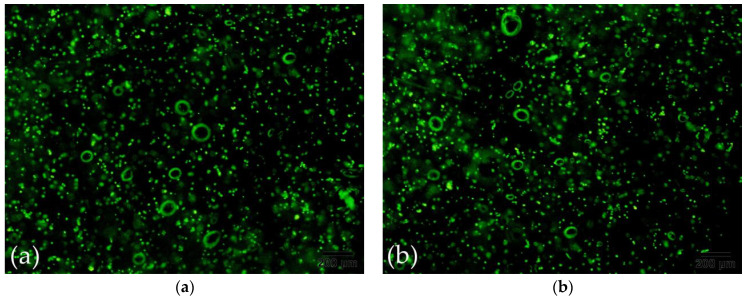
Cell survival states in the gelatin/alginate/agarose hydrogel under different extrusion pressures. (**a**) Under a pressure of 0 MPa. (**b**) Under a pressure of 0.2 MPa. (**c**) Under a pressure of 0.4 MPa. (**d**) Under a pressure of 0.5 MPa.

**Figure 8 polymers-13-04028-f008:**
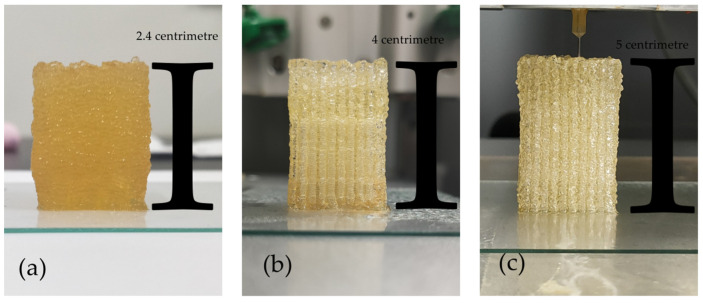
Structures constructed with different concentrations of agarose: (**a**) 0%, (**b**) 0.2%, (**c**) 0.4%.

**Figure 9 polymers-13-04028-f009:**
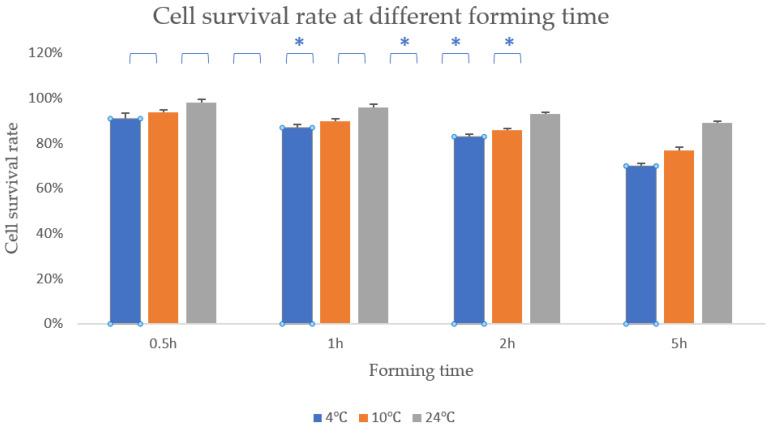
Cell survival rate at different forming temperature. * means that there are statistical significances between the two adjacent temperature.

**Figure 10 polymers-13-04028-f010:**
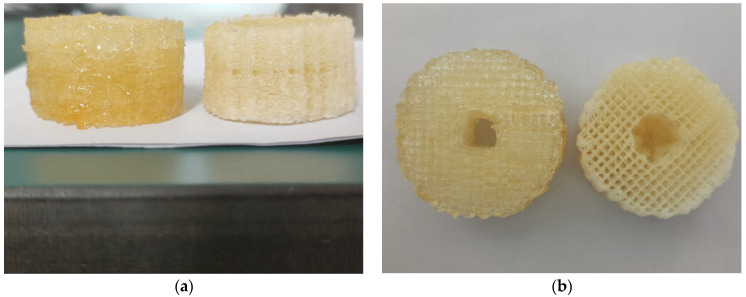
Non-temperature-controlled bioprinting (left) and semi-temperature-controlled bioprinting (right) blood vessel-like morphology: (**a**) front view, (**b**) top view.

**Figure 11 polymers-13-04028-f011:**
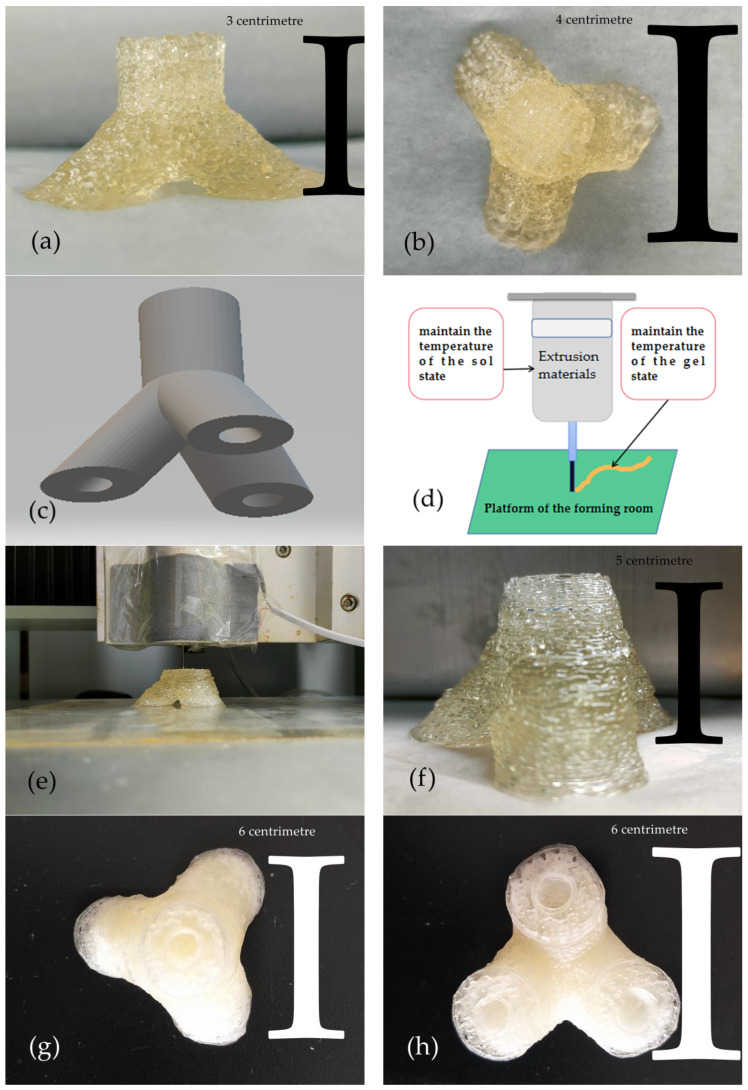
(**a**) Front view of solid structure. (**b**) Top view of solid structure. (**c**) Hollow trigeminal vascular structure model. (**d**) Schematic diagram of extrusion printing. (**e**) A shape view of the hollow, trigeminal vascular structure with a heating plate. (**f**) Front view of the hollow trigeminal vascular structure. (**g**) Top view of the hollow trigeminal vascular structure. (**h**) Bottom view of the hollow trigeminal vascular structure.

**Figure 12 polymers-13-04028-f012:**
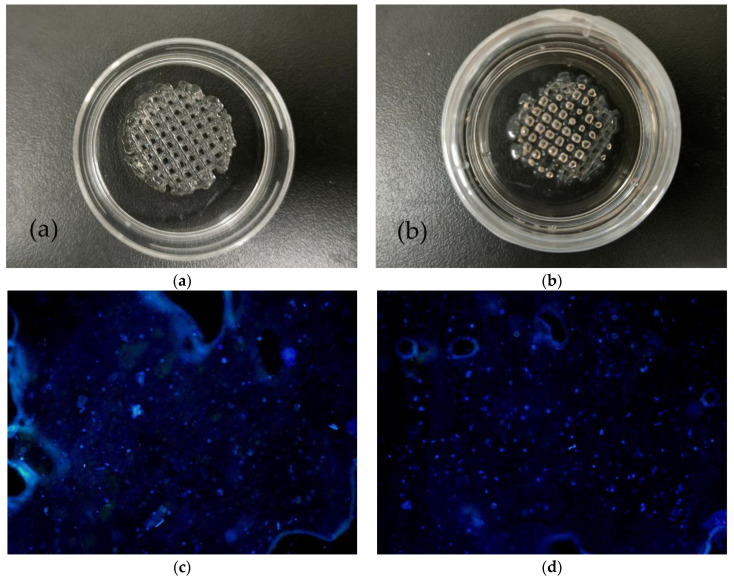
(**a**) The shaped structure with cells. (**b**) Cell-containing structure added to culture medium. (**c**) Cell proliferation on the first day of culture. (**d**) Cell proliferation on the second day of culture. (**e**) Cell proliferation on the fourth day of culture. (**f**) Cell proliferation on the sixth day of culture.

**Figure 13 polymers-13-04028-f013:**
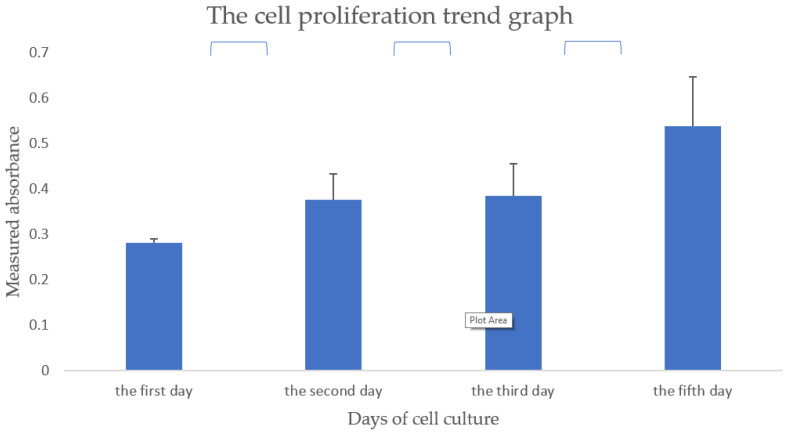
Cell proliferation trend graph of hydrogel composed of 15% gelatin, 2% sodium alginate, and 0.2% agarose.

**Table 1 polymers-13-04028-t001:** Main parameters of three printing procedures.

Main Parameter	Traditional Printing	Semi-Temperature Controlled	Non-Temperature Controlled
Formation temperature (T)	4 °C	10 °C	20–30 °C(room temperature)
Layer thickness (Δh)	0.55 mm	0.3 mm	0.40 mm
Nozzle scanning speed (V_s_)	3 mm/s	3.5 mm/s	2 mm/s
Material extrusion speed (V_n_)	0.1 mm/s	0.08 mm/s	0.1 mm/s
Inner diameter of nozzle (D)	0.40 mm	0.30 mm	0.35 mm
Grid width	2.2 mm	2.2 mm	2.2 mm
Material filling angle	0°, −90°	0°, −90°	0°, −90°

**Table 2 polymers-13-04028-t002:** Concentrations of polymer components.

Main Parameter	The Average Matrix Wall Thickness (µm)	The Average Pore Diameter (µm)	The Porosity (%)
10% gelatin, 2% alginate, 0.2% agarose	20 ± 10	150 ± 20	70–80%
15% gelatin, 2% alginate, 0.2% agarose	40 ± 10	125 ± 20	50–60%
20% gelatin, 2% alginate, 0.2% agarose	60 ± 10	100 ± 20	30–40%
5% gelatin, 0% alginate, 0.2% agarose	10 ± 5	60 ± 20	80–90%
5% gelatin, 4% alginate, 0.2% agarose	20 ± 5	120 ± 20	70–80%
5% gelatin, 8% alginate, 0.2% agarose	30 ± 5	180 ± 20	40–50%
5% gelatin, 2% alginate, 0% agarose	10 ± 5	60 ± 20	70–80%
5% gelatin, 2% alginate, 0.4% agarose	15 ± 5	100 ± 20	80–90%
5% gelatin, 2% alginate, 0.8% agarose	20 ± 5	100 ± 20	80–90%

**Table 3 polymers-13-04028-t003:** Extrusion pressure changes under different printing times.

Printing Time (min)	Extrusion Pressure (Non-Temperature-Controlled) (MPa)	Extrusion Pressure (Semi-Temperature-Controlled) (MPa)	Extrusion Pressure (Temperature-Controlled) (MPa)
0	0.01 ± 0.05	0.04 ± 0.05	0.09 ± 0.05
2.33	0.07 ± 0.05	0.11 ± 0.05	0.18 ± 0.05
4.66	0.13 ± 0.05	0.18 ± 0.05	0.27 ± 0.05
7	0.19 ± 0.05	0.25 ± 0.05	0.36 ± 0.05
9.33	0.25 ± 0.05	0.32 ± 0.05	0.445 ± 0.05
11.66	0.295 ± 0.05	0.375 ± 0.05	0.51 ± 0.05
14	0.31 ± 0.05	0.405 ± 0.05	0.565 ± 0.05
Average value	0.179 ± 0.05	0.24 ± 0.05	0.346 ± 0.05

**Table 4 polymers-13-04028-t004:** The cell death rate under different extrusion pressures.

Cell Death Rate under 0 MPa (%)	Cell Death Rate under 0.2 MPa (%)	Cell Death Rate under 0.4 MPa (%)	Cell Death Rate under 0.5 MPa (%)
0–5%	5–10%	10–20%	20–30%

**Table 5 polymers-13-04028-t005:** The cell proliferation rate in different periods.

Cell Proliferation Rate on the First Day (%)	Cell Proliferation Rate on the Second Day (%)	Cell Proliferation Rate on the Fourth Day (%)	Cell Proliferation Rate on the Sixth Day (%)
10–20%	25–35%	50–70%	90–120%

## Data Availability

All the data is true and reliable, and anyone can send an email to the author’s email to request the original data.

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
