# Peer review of "The Effect of Agarose on 3D Bioprinting"

_polymers, 2021, doi:10.3390/polym13224028_

Round 1

Reviewer 1 Report

The paper already reported interesting results and had a very clear structure. In addition, the revised version of the work is suitably improved also stressing the concept of shear thinning which plays a crucial role in the additive manufacturing processes based on injection/extrusion method .

However, even though shear thinning behavior assessed by steady state shear measurements (viscosity as function of shear rate) suggest the possibility to inject the material through a needle, functional injectability tests clearly demonstrate the injectability through needles with specific diameter providing information on the maximum load (and, hence, pressure) involved in the process, which should be consequently optimized according to the obtained results. In addition, the viscoelastic properties before and after the injection through the employed nozzles are important for the process and for the final properties of the fabricated devices (e.g., the injection through nozzles may strongly alter the material properties). If the authors did not perform such analyses, I suggest to briefly stress these features/concepts using some sentences, also without reporting any references. Such concepts are important in the design of devices according the a specific technology.

Author Response

Thank you for your objective and fair evaluation. Your suggestions have greatly improved the quality of my manuscript. I have revised my manuscript according to your suggestions. Thank you for your review.

Reviewer 2 Report

The authors sincerely provided replies to some critical issues raised in the previous review stage.

It is considered that the present version of the manuscript would be acceptable. 

Author Response

Thank you for your objective and fair evaluation. And thank you for spending a lot of time reviewing my manuscript.

This manuscript is a resubmission of an earlier submission. The following is a list of the peer review reports and author responses from that submission.

Round 1

Reviewer 1 Report

Refer to reviewer report

Reviewer 2 Report

The paper presents a so-called "pregelatinization effect of agarose" to enhance bioprinting. In many ways, the paper results are very inaccurate and not trustable. It is hard to believe that the standard deviations of all the experimental groups from Figures 3, 4 and 7 have the same value. Apart from these concerns, Figures 8 and 13 are wrong. The cell survival stains everything in green and the (Figure 8), which has no sense; and no red dots were found. In Figure 13, I totally disagree with the text included in the Section Results: 

"As presented in Figure 13(d), After two days of culture,
632 observation under a fluorescence microscope can clearly show that the
633 number of cells has increased slightly, which indicates that the
634 material has better biocompatibility. After culturing for four days, the
635 number of cells has increased to a certain extent as presented in Figure
636 13(e). Time went by little by little. On the sixth day, the cells had
637 spread all over the area under the fluorescence microscope as
638 presented in Figure 13(f)."

My recommendation is to reject the article.

Reviewer 3 Report

Overall comments:

The subject matter of the paper dealt with the development of two 3D bioprinting methods: agarose semi-controlled temperature bioprinting and agarose non-controlled temperature bioprinting by using the physical cross-linking of agarose to replace the pregelatinization effect of gelatin.

The present study is worth investigating and the manuscript itself is considered to be theoretically and structurally reasonable. However, this manuscript lacks key in-depth discussion in which the present data should be compared with those obtained in similar systems or others related to them rather than only simple description of the results. Furthermore, the interpretations of results are not reasonable and extremely arbitrary and unsophisticated. Thus, they should more elaborate upon what is of paramount importance together with general significance. I worry that this masks what appears to be very important subject. Together with this major point, there are some specific concerns that should be addressed in a point-by-point manner. If such major and minor issues were all cleared by the authors, this paper can be re-evaluated.

Specific concerns:

1) Figure 1 doesn't make any meaning, so it would be better to delete it.

2) Figure 9: Live/dead assay results seem inappropriate. Considering that the green signal is prevalent in the background, it is suspicious that the signal of green channel is increased compared to red channel. In addition, the morphology of cells cannot be confirmed due to low magnification, which can be interpreted that the dyes simply aggregated with cell debris or the green signal was from autofluorescence of gels. Moreover, it is hard to compare cell viability with the provided images. To be convinced more, provide the data to quantify the number of live and dead cells.

3) Figure 15: It is difficult to assess cell proliferation with the provided DAPI staining. Also, cell population is not significantly increased during culture. Provide more convincing data through CCK-8 assay and proliferation marker staining such as ki-67.

4) Provide the reason why glycerol was used despite high cytotoxicity.

5) In general, high viscous hydrogels can exhibit good printability, but they do not promote essential requirements of cell growth such as cell spreading, proliferation, nutrients supply, and metabolites removal. Therefore, many studies use low concentrations of hydrogels for facilitation of cell growth and the insufficient mechanical strength has been the major bottleneck. In this study, the incorporation of agarose into gelatin induced high viscous bioink and cell affinity was evaluated through various experiments. However, experimental design and data interpretation are considered to be neither sophisticated nor reasonable.

Reviewer 4 Report

- The approach is interesting and the topic is appropriate for the journal.

  • The work has a very clear structure and all the sections are well written in a way that is easy to read and understand.
  • However, some modifications and improvements are needed to enhance the quality of the paper.

  • The paper is focused on the effect of agarose on 3D bioprinting , reporting interesting results. In the “Introduction” section, the authors start to discuss about tissue engineering and 3D bioprinting technologies. Even though the authors already report some strategies in literature related to 3D bioprinting as well as to gels/hydrogels, I also suggest to BRIEFLY introduce the last progresses in the design and analysis gels/hydrogels with tailored properties and integrated functionalities which should be processed using 3D bioprinting technologies starting from the materials or from the reactive solutions. As the additive manufacturing processes based on injection/extrusion method  are strongly influenced by rheological properties and functional injectability of the materials/starting solutions, in the introduction I suggest to briefly stress the important role of the rheological properties (G’ and G’’ before and after the injection through nozzles, viscosity as function of shear rate – shear thinning is also cited on page 17) and functional injectability through needles when processing gels/hydrogels and reactive solutions (e.g., Connective Tissue Research, 2020, 61(2), pp. 152–162…). Then  the authors should continue to stress their study related to the effect of agarose on 3D bioprinting. All of this should improve the quality of the paper, reporting important features as well as further design methodologies  and analysis of gels/hydrogels and reactive solutions, thus helping the different kinds of readers to better understand the value of their work and the importance of some technical features.
  • The Introduction and/or discussion section as well as the list of references should be improved according to the above reported comments.
  • In figure 8, on the y-axis “Mpa” should be replaced by “MPa”.
  • On page 17, “Mpa” should be replaced by “MPa”. I suggest to check this feature in all the manuscript.
  • With regard to the hardness test results, I suggest to better explain the expression “Hardness (OO)” for the potential readers. As an example, It is well known that ASTM D2240-00 testing standard calls for different scales. Accordingly, this should be better explained.
  • The quality of some figures should be improved.
  • The title is adequate and appropriate for the content of the article.
  • The abstract contains information of the article.
  • Figures and captions are essential and clearly reported.

Round 2

Reviewer 2 Report

After reviewing the authors' corrections and the new information provided, this so-called "pregelatinization effect of agarose" paper does not show any proof that it can enhance bioprinting. Also, the paper results continue to show inaccuracy and I do not trust them. My recommendation is still to reject the work.

Reviewer 3 Report

The authors sincerely provided replies to some critical issues raised in the previous review stage.

It is considered that the present version of the manuscript was sufficiently well revised according to the reviewers' comments.

This manuscript would be acceptable if the authors could clear specific issues in the following:

1) There are missing some marks for statistically significant differences in Figs. 3, 7, 10, and 14.

2) Please analyze the images of Figs. 8 and 13 (c)-(f) quantitatively provide them as graphs to relatively compare each other.

Author Response

Dear reviewer,

Thank you very much for spending a lot of time reviewing my manuscript. Your fair and objective evaluation has benefited me immensely. I have revised my manuscript according to the two comments you put forward. I hope the revised manuscript can meet your requirements.
We look forward to your reply.

Best wishes,

Chi Gong